# A hydrogel model of the human blood-brain barrier using differentiated stem cells

**Nandita Rahatekar Singh[1], Radka Gromnicova[1], Andreas Brachner[2], Igor Kraev[1], Ignacio A. Romero[1], Winfried Neuhaus[2,3], David Male[1]***

**1** Department of Life, Health and Chemical Sciences, The Open University, Milton Keynes, United Kingdom, **2** Competence Unit Molecular Diagnostics, Center Health and Bioresources, AIT—Austrian Institute of Technology GmbH, Vienna, Austria, **3** Department of Medicine, Danube Private University, Krems, Austria

* david.male@open.ac.uk

**Data Availability Statement:** All relevant data are within the paper and its Supporting Information files.

## Abstract

An *in vitro* model of the human blood-brain barrier was developed, based on a collagen hydrogel containing astrocytes, overlaid with a monolayer of endothelium, differentiated from human induced pluripotent stem cells (hiPSCs). The model was set up in transwell filters allowing sampling from apical and basal compartments. The endothelial monolayer had transendothelial electrical resistance (TEER) values >700Ω.cm$^2$ and expressed tight-junction markers, including claudin-5. After differentiation of hiPSCs the endothelial-like cells expressed VE-cadherin (*CDH5*) and von-Willebrand factor (*VWF*) as determined by immunofluorescence. However, electron microscopy indicated that at set-up (day 8 of differentiation), the endothelial-like cells still retained some features of the stem cells, and appeared immature, in comparison with primary brain endothelium or brain endothelium *in vivo*. Monitoring showed that the TEER declined gradually over 10 days, and transport studies were best carried out in a time window 24-72hrs after establishment of the model. Transport studies indicated low permeability to paracellular tracers and functional activity of P-glycoprotein (*ABCB1*) and active transcytosis of polypeptides via the transferrin receptor (*TFR1*).

## 1. Introduction

In the past 30 years several *in vitro* models of the blood-brain barrier (BBB) have been developed, which are based on primary brain endothelial cells or cell lines [1]. These models allowed partial replacement of experimental animals, when measuring transport of therapeutic agents, biomolecules and nanocarriers into the CNS. However, since the solute transporters (SLCs) and some of the multi-drug (ABC) transporters present on brain endothelium differ in expression between species [2–4], it is important to use human cells, when modelling transport in humans. Primary human brain endothelium is a scarce resource, derived from normal tissue removed during elective surgical resections. Moreover, the phenotype of primary human brain endothelium decays *in vitro* with successive passages [5]. For these reasons earlier *in vitro* models of the human BBB generally used immortalised cell lines, such as hCMEC/D3 or

**Funding:** This project was funded by the Innovative Medicines Initiative 2, Joint Undertaking (JU) under grant agreement No. 807015, (IM2PACT - Identifying Models and Mechanisms Predicting Access of Therapeutics into the Brain.), to DM, IR and WN. The JU receives support from the European Union's Horizon 2020 research and innovation programme and EFPIA. https://www.imi.europa.eu/ The funders had no role in study design, data collection and analysis, decision to publish, or preparation of the manuscript.

**Competing interests:** The authors have declared that no competing interests exist

HBMEC/ciβ, which express many of the receptors present on primary brain endothelium while maintaining phenotypic stability over many passages [6, 7].

*In vivo*, brain endothelium has a trans-endothelial electrical resistance (TEER) >1000 $\Omega$. cm$^2$ [8] and the cells effectively prevent paracellular diffusion of molecules with molecular weight >1kDa. Hence, desirable characteristics of a BBB model are the expression of continuous tight junctions, a high TEER and corresponding low permeability for paracellular tracers (e.g., lucifer yellow, sodium fluorescein, FITC-dextran). Tight junctions can be identified by expression of ZO-1 and occludin, together with cell-type specific members of the claudin family. Claudins -1. -3, -5 and -12 are present in endothelial tight junctions; claudin-5 is particularly highly expressed on brain endothelium and often used as a marker of these cells [9]. All vascular endothelial cells also have adherens junctions with VE-cadherin (cadherin-5); in addition, brain endothelium selectively expresses high levels of cadherin-10 [10]. Immortalised brain endothelial cell lines express these junctional molecules, but they are less well organised than in primary cells or endothelium *in vivo*. The weak organisation of the tight junctions results in low TEER (<100 $\Omega$.cm$^2$) and relatively high permeability for serum proteins and other biomolecules. Consequently BBB-models using immortalised brain endothelial cell lines have to be evaluated carefully for the contribution of paracellular and transcellular routes when examining transport of therapeutic agents and biomolecules with molecular weight <10kDa, since higher paracellular leakage might lead to an overestimation of drug permeability [11].

Other desirable characteristics of a BBB-model are the presence and functional activity of multi-drug resistance transporters (e.g., ABCB1, ABCG2, ABCC1, 3, 4 and 5) [12] and solute transporters such as the glucose transporter GLUT1 (SLC2A1) and amino acid transporters (eg SLC7A5) [13]. Another important group of cell-surface markers are involved in the transport of specific proteins and lipoproteins, including the transferrin receptor (TFR1), the insulin receptor (INSR) and the LDL receptor (LRP1). These transporters are selectively expressed at high levels on brain endothelium, in comparison with endothelium from other tissues [4]. They are also potentially important as they present targets for transport of larger therapeutic agents which can cross the endothelium by vesicular uptake and receptor-mediated transcytosis (RMT) [14].

To address the constraints of immortalised lines and the limited supply of primary brain endothelium, several methods have been developed to differentiate brain endothelium from human inducible pluripotent stem cells (hiPSCs) [15, 16]. hiPSCs have some transporters and tight junction proteins in common with brain endothelium (eg ZO-1), but the overall composition of their junctional proteins and transporters is different. It is therefore essential to check that 'brain endothelium' derived from differentiated hiPSCs expresses true endothelial markers such as von-Willebrand factor (VWF), PECAM (CD31) and VE-cadherin (*CDH5*, CD144). In addition, differentiation should result in increased expression of brain endothelium specific markers such as claudin-5 (*CLDN5*) and cadherin-10 (*CDH10*) [9, 10]

We have previously developed a hydrogel model of the blood-brain barrier, which includes astrocytes or other glial cells incorporated into a collagen gel, overlaid with a monolayer of hCMEC/D3 cells [17]. This model has advantages over endothelium on transwell filters, in that material applied to the apical surface of the endothelium can be released at any point on the basal surface of the cells; it does not become trapped between the cells and the filter and can diffuse to target cells within the gel or to the basal chamber of the transwell. Different types of glial cell can be incorporated into the hydrogel, to determine whether molecules that have crossed the endothelium are taken up by glia or can activate them. The model has also been used to examine lymphocyte trans-endothelial migration, and transcytosis of gold glyconanoparticles. However, it is not suitable for measuring movement of molecules <10kDa

because the junctions between the hCMEC/D3 cells are not as tight as in brain endothelium *in vivo* (TEER <50 $\Omega$.cm$^2$), allowing leakage by the paracellular route.

Recently, several newer hydrogel models of the blood-brain barrier have been developed, reviewed in [18]. The aim of the research described here was to replace low TEER hCMEC/D3 cells in our previously described hydrogel model with high-TEER, hiPSC-derived brain endothelium. Following on, the objectives were to check the phenotype of the differentiated endothelium and measure functionality of the model using fluorescent tracers for junctional integrity and substrates which are normally transported or excluded by human brain endothelium.

## 2. Materials and methods

### 2.1 Differentiation of induced pluripotent cells to brain endothelial-like cells

Human induced pluripotent stem cells (SBAD-02-01) were kindly supplied Dr Zameel Cader, University of Oxford. Expansion and differentiation of hiPSCs was done on Matrigel (BD Biosciences) coated 6-well plates (Nunc) in mTeSR1 medium (STEMCELL Technologies). Differentiation of SBAD-02-01 was performed as described [15]. Briefly, cells were thawed from the frozen vial and were expanded until 70% confluent on a Matrigel coated 6 well plate. After expansion, cells were passaged by use of accutase solution (Sigma-Aldrich) and seeded at 7.5x10$^3$ hiPSC/cm$^2$ on Matrigel coated 6-well plates. Cells were further expanded in mTeSR1 medium. After achieving an optimal cell density of 2.5 x10$^4$–3.5 x10$^4$ hiPSC/cm$^2$, the cells were shifted to unconditioned media (UC) for 6 days. UC medium consisted of 78.5% DMEM/F-12 + 20% KnockOutTM Serum Replacement (Life Technologies) + 1% MEM NEAA (Sigma-Aldrich) + 1mM L-glutamine (Sigma-Aldrich) + 0.1 mM β-mercaptoethanol (Sigma-Aldrich). The medium was changed completely every day. Cells were then cultured for 2 days in human Endothelial-SFM (Life Technologies), containing 0.5% B27$^{TM}$ (50x) (Thermo Fisher Scientific) supplemented with 20 ng/mL hbFGF (PeproTech) and 10 μM all-trans Retinoic Acid (Sigma-Aldrich), called EC medium + hbFGF + RA.

### 2.2 Cell selection before seeding

It was found necessary to isolate a high-adherence population of cells from the differentiating cells, as unfractioned cells could detach from the hydrogel-substrate, leaving gaps in the monolayers, resulting in unusable cultures. On day 8 after shift to UC medium, the cells were detached with accutase (45 min, 37˚C) and resuspended in EC medium. They were seeded onto collagen- and fibronectin-coated 6-well plates (Nunc) prepared one day in advance. These adherence plates were coated with 1ml per well of HBSS containing 16μg/ml type-IV collagen (Sigma-Aldrich) and 4μg/ml fibronectin (Santa Cruz). One adhesion plate was used for each plate of differentiated hiPSCs.

The cells were incubated for 60 min at 37˚C on the adhesion plates and weakly-adherent cells were removed by aspiration followed by two washes with HBSS at 37˚C. Strongly adherent cells were released with accutase, and these cells were used for further seeding onto hydrogels or transwell inserts.

### 2.3 Transwell insert preparation and cell seeding

For simple cultures of endothelial cells (no hydrogel) 12-well transwell inserts (1cm$^2$) with a pore size of 0.4 microns (Greiner- bio) were used. The inserts were treated with 100μl of collagen type-IV and fibronectin in water at a final concentration of 0.4mg/ml collagen-IV: 0.1 mg/

ml fibronectin. The coated transwell inserts were incubated at 37°C overnight, residual substrate removed, and the inserts washed x2 in HBSS before use. The strongly adherent cells, obtained on day8 were seeded at an optimized cell density of $1x10^6$ cells per filter in 700μl EC medium, with B27 + hbFGF + RA for 24 hrs. Medium was exchanged after 24 hrs with EC medium without hbFGF and RA.

## 2.4 Preparation of hydrogel model

The collagen hydrogels were formed in 12-well transwell inserts. Rat tail type-I collagen (First Link) with a concentration of 5mg/ml was mixed with 10xMEM to produce an isotonic solution and then neutralised using 1M sodium hydroxide (assessed by colour change of the phenol red indicator). In experiments where astrocytes were incorporated into the gel, the neutralised collagen was added to the cell suspension at this stage and mixed to ensure even distribution of cells before transferring to the transwell inserts for gelation, as previously described [17]. Primary human astrocytes at passage 2–4 (ScienCell) were cultured in human astrocyte medium containing 2% foetal bovine serum and recommended supplements (ScienCell). They were incorporated into the hydrogels at densities up to $2 x 10^5$/ml as previously described [17]. Approximately 300μl of the gel solution was added to each transwell to produce a gel of ~3mm thickness. Gelation took approximately 10 min at 37°C. The gels were then coated with a mixture of collagen-IV and fibronectin as described above and kept overnight in the incubator before being washed with HBSS and finally seeded with $1 x 10^6$ highly-adherent differentiated endothelial-like cells per insert. Fig 1 shows the time-course of differentiation of the hiPSCs and the layout of the hydrogel model.

## 2.5 Measurement of trans endothelial electrical resistance using Cellzscope and Endohm meter

A Cellzscope device (Nanoanalytics version 3.05-Plus) was used to record continuous TEER measurements to standardize the model and the results were compared with individual time-point results from an Endohm meter, EVOM2 from World Precision Instruments. The medium was changed completely every day and TEER measurements were made 24 hr following media exchange on day 10 after differentiation using the Endohm meter and the Cellzscope with 1 cm diameter electrodes and standard spectrum settings: frequency 1 Hz-100 kHz, points per decade 9 and logarithmic spacing. The time-course of the TEER readings on the Cellzscope extended up to 10 days and required temporary removal of the transwells from the Cellzscope to exchange media. TEER values for the empty inserts are automatically subtracted from the TEER values for the inserts with cells and cells on hydrogels to obtain final TEER values in $\Omega.cm^2$. The Endohm meter was used to take measurements before and after the start of all transport studies. All readings were collected in triplicate. For transport assays, only hydrogel cultures with cells that showed TEER $>200$ $\Omega.cm^2$ on the Endohm meter before and after the assay were included as valid results.

## 2.6 Immunofluorescence staining

All steps for immunofluorescence were performed at room temperature. After differentiation of the cells at day 8, samples of the cells were placed in Labtek chamber slides (Thermofisher catalogue number 177445) coated with collagen IV/fibronectin as described above. After 48 hours in EC medium the cells were washed with PBS and fixed with 4% paraformaldehyde for 10 minutes. After washing twice with PBS, cells were permeabilized for 5 minutes with 0.2% Triton X-100 (Sigma-Aldrich) in PBS. Cells were washed three times for 5 minutes with PBS and then blocking was done for 30 minutes in 100mg/ml bovine serum albumin (Sigma

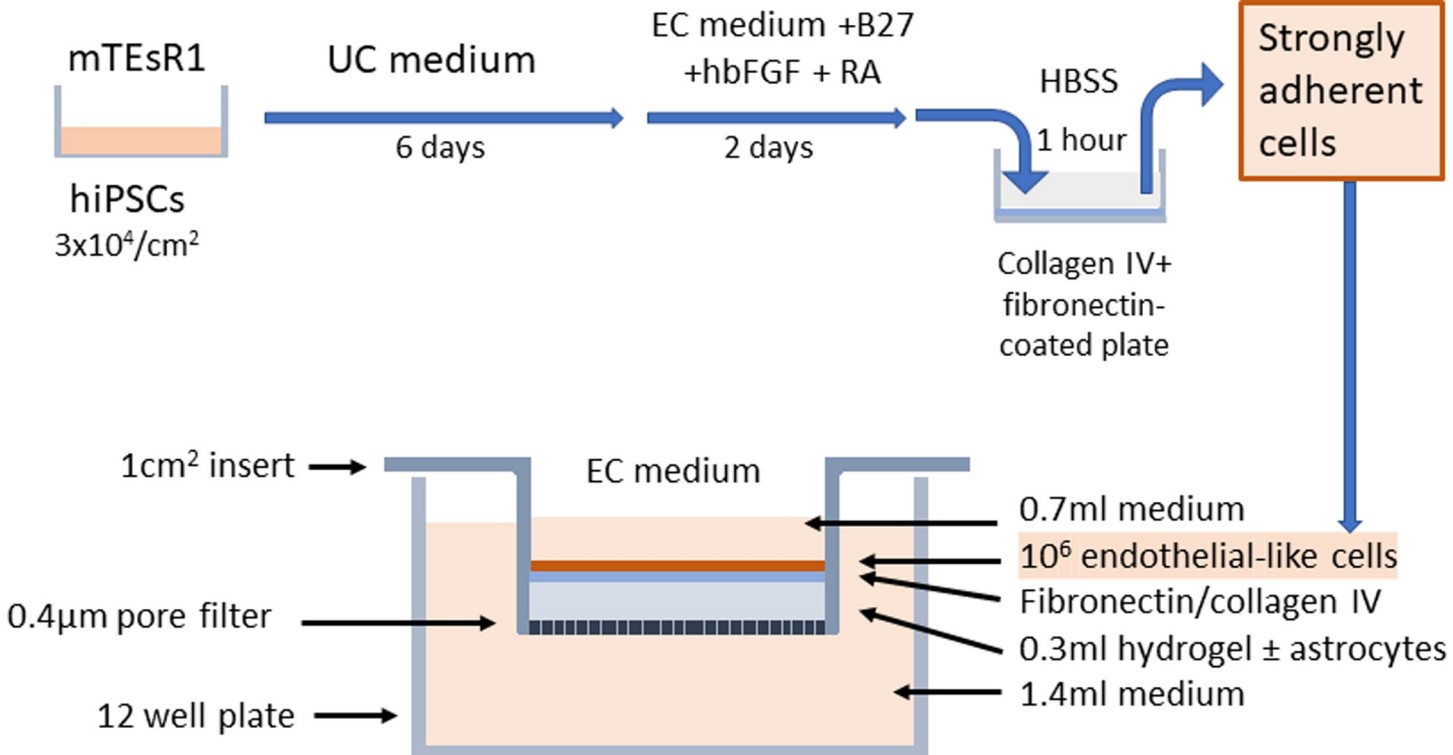

**Fig 1. Time course of the differentiation of the hiPSCs and a schematic representation of the hydrogel model on transwell filters.** HiPSCs were maintained in mTeSR1 medium. When they had reached an optimum density of $3x10^4/cm^2$ they were switched to unconditioned (UC) medium for 6 days, followed by serum free endothelial cell (EC) medium containing B27, human basic fibroblast growth factor (hbFGF) and retinoic acid (RA) for 2 days. Strongly adherent cells were isolated by 1hour incubation on collagen IV/fibronectin-coated plates in Hanks balanced salt solution (HBSS). Adherent, endothelial-like cells were plated onto a $1cm^2$ transwell insert that had a 0.3ml collagen I hydrogel and a fibronectin/collagen IV surface layer, prepared 24hours previously. The cells were cultured in EC medium for 24-48hours before use, and the inserts were usable for up to 7 days.

Aldrich). Cells were then incubated with the primary antibody diluted in the blocking solution at 4˚C overnight. After washing for three minutes in PBS the cells were treated with 10μg/ml goat anti-rabbit IgG H&L Alexa Fluor 488 (Abcam, ab 150077), which was diluted in 20mg/ml BSA and incubated for 1 hour at room temperature. Table 1 shows the list of primary antibodies used. After three additional washing steps (5 minutes each) cells were covered with a drop of vector shield (eBioscience) and mounted with a cover slip. The cells were visualised with a Zeiss Axio Observer 7 or a Leica TCP SP5 confocal fluorescence microscope (VWF-stained).

**Table 1. Primary antibodies.**

| Antibody specificity | Catalogue number | Species | Supplier | Final concentration |
|---|---|---|---|---|
| ZO-1 | 21773-1-AP | Rabbit | Proteintech | 2μg/ml |
| Claudin 5 | 34–1600 | Rabbit | Abcam | 2μg/ml |
| Occludin | 40–4700 | Rabbit / IgG | Invitrogen | 2μg/ml |
| VE-cadherin | Ab 33168 | Rabbit | Abcam | 2μg/ml |
| CD-31 * | Ab28364 | Rabbit | Abcam | 1:50 |
| VWF | F3520 | Rabbit | Sigma | 2μg/ml |

* Glutaraldehyde fixation

## 2.7 Electron microscopy

The cells on collagen gels were fixed in 2.5% glutaraldehyde for 2 hrs at room temperature. They were then washed in 0.1M Phosphate buffer 3x and then post-fixed in 1% osmium tetroxide (in 0.1M Phosphate buffer) overnight at room temperature. The cultures were dehydrated using a series of acetones, namely 30% for 10 min, 50% for 10 min, 70% for 20 min, 100% for 20 min twice, and 100% sieved (with a molecular sieve) for 20 min. The samples were then incubated in mixture of 100% acetone and epoxy resin at 1:1 ratio overnight at room temperature. The epoxy resin infiltration was performed by replacing acetone/epoxy mixture with pure epoxy resin. The samples were incubated in epoxy resin for 2 hrs at room temperature, and another change of epoxy resin was made, with further 2 hr incubation. The samples were embedded into fresh epoxy resin in a mould and polymerized into blocks at 60°C for 48 hrs. The blocks were then micro-sectioned at 80 nm and sections were collected onto a pioloform-filmed copper slot grid. The sections were counter-stained using 3% aqueous uranyl acetate for 30 min and Reynold's lead citrate for 10 min. Imaging was performed in a JEM 1400 electron microscope (JEOL) using an AMT XR60 camera (Deben, UK), at an acceleration voltage of 80 kV and magnifications between 2k and 50k.

## 2.8 Barrier functionality

Barrier function across the monolayer was calculated using 70kDa FITC dextran (Dx) and lucifer yellow (Ly). Both assays were conducted 2 days after setting up the filters and were done with four replicates. Both assays were performed twice with different sets of differentiated cells. The controls were transwell filters alone (blank filter), filters with hydrogel (no cells) and filters with endothelial-like cells (no hydrogel). Stock solutions of Ly and Dx-were prepared at 2mg/ml in HBSS without phenol red. The TEER value was measured using an Endohm meter before the start of the assays. From the top chamber, containing 700μl medium, 350μl of EC media was replaced by 350μl of stock tracer solution to bring the final concentration to 1mg/ml of Dx or Ly. From the lower chamber of the transwell, containing 1.5mls of medium, 100μl samples were collected at regular intervals and transferred to a 96 well black-wall plate. The samples were replaced by 100μl of EC media containing B27 only. TEER values were again recorded using the Endohm meter at the end of the transport assay. After the completion of the assay the inserts were placed in a Cellzscope to monitor for any drop in TEER over a period of 3–4 days.

## 2.9 Transport assays

P-glycoprotein (Pgp = ABCB1) and breast cancer resistance protein (BCRP = ABCG2) are the major efflux transporters of the blood brain barrier. Functional activity of ABCB1 was determined by the rate of transport of a fluorescent Pgp-substrate, 5μM Rhodamine-123 (Sigma) in the presence or absence of the Pgp-inhibitor, 0.4μM Zosuquidar (Sigma). The activity of ABCG2 was determined by the rate of transport of a fluorescent substrate, 10μM mitoxantrone (Sigma), in the presence or absence of the BCRP-inhibitor, 1μM Ko143 (Sigma). Working solutions of the substrates and inhibitors were prepared in HBSS at 10x the stated final concentrations. One hour before the transport assay 70μl of (10x) inhibitor was added to the upper chamber of the transwells and 150μl to the lower chamber and swirled to mix. At the start of the transport assay 70μl of medium was removed from the upper chamber and replaced with 70μl (10x) substrate. Samples of 100μl were taken from the lower chamber over a period of 0-24hrs and transferred to a black-wall 96 well plate for fluorimetry. The sample was replaced by 100μl of EC media with B27. Rhodamine-123 fluorescence was measured with excitation 510nm and emission at 560nm. Mitoxantrone fluorescence was measured with excitation

620nm and emission at 680nm after the addition of 10% SDS to a final concentration of 1% SDS. Standard curves of the substrates in EC medium were prepared on the same 96-well plates and read under the same conditions as the test samples to allow quantitation of the amount of tracer transferred to the lower chamber.

## 2.10 Functional transferrin receptors

The transport of peptides that bind the transferrin receptor was measured on hydrogel cultures with a TEER >200 $\Omega.cm^2$. Each assay included four replicates (cells on hydrogels), and controls with cells only, filter only and gel only. The peptides included three which bind strongly to the TfR (pep-1, pep-10, pep-10M) one that binds weakly (pep-2) and a control peptide that does not bind the TfR (pep-R1) [19]. All peptides carried a fluorescent tag (FITC) on the N-terminal amino acid. Stocks of fluorescent peptides were prepared and stored at 10mg/ml in DMSO. Working solutions were made at 250µg/ml in HBSS and 1/10 volume added to the apical chamber of the hydrogel cultures to produce a final concentration of 25µg/ml in the apical chamber. Samples were collected at regular intervals over 0–6 hrs in a 96 well black-wall plate and readings were taken at an excitation of 485nm, emission 544nm.

## 2.11 Analysis of permeability and transport assays

Transport of fluorescent tracers, peptides or substrates across the hydrogel cultures was determined by first establishing a time period when the rate of transport was linear. In hydrogels there was generally a lag-period of approximately 1hr, before any tracers could be detected in the lower chamber, due to the length of the diffusion path through the hydrogel. For Ly, Dx, TfR-binding peptides and Rho-123, linear transport occurred at 1–6 hrs and the concentration of fluorescent tracers or substrates in the lower chamber was derived by interpolation from standard curves in EC medium. Mitoxantrone could only be reliably detected at 24hrs, and results are shown for the 24hr time-point. For paracellular permeability studies (Ly, Dx), the cleared volume and permeability Pe, were calculated according to previously described methods [20]. The values obtained with the hydrogels were compared with cells on filters (no hydrogel) or hydrogels on filters (no cells) and the transport rate per unit time, calculated. For TfR-transport assays, the rate of accumulation in the lower compartment was compared for different TfR-binding peptides. For ABC-transporters (ABCB1, ABCG2) the rate of accumulation of the substrate in the lower compartment was compared with or without the appropriate inhibitor.

## 2.12 Gene expression profile of differentiated brain endothelial-like cells

RNA was extracted from the differentiated brain endothelial-like cells at day 8 after transfer to UC medium, to yield RNA from 1. Unfractionated cells, 2. Adherent cells, 3. Non-adherent cells using the Qiagen AllPrep DNA/RNA/Protein Mini Kit (Qiagen, UK). For gene expression analysis a high-throughput qPCR barrier chip was applied as previously reported [21]. In detail, 20 µl cDNA was generated from 250 ng RNA with the High-Capacity cDNA Reverse Transcriptase Kit (Thermo Fisher Scientific). After preamplification of the targets in the samples, the high-throughput qPCR chip was performed with the preamplified cDNA in 96x96 chips applying the Biomark™ system (Fluidigm®). Gene expression was normalised for each analysis against the recommended house-keeping gene (PPIA) (ΔCT) before comparison of expression of individual genes in different RNA preparations from the same experiment (ΔΔCT), which is expressed as relative level of expression against the control sample (= 1).

### 2.13 Statistical analysis

Data for transport of TfR-binding peptides or substrates were compared by one way ANOVA followed if appropriate by unpaired t-test (substrates) or Tukeys post hoc test (peptides). Data for Rho-123 or mitoxantrone transport compared values with and without the appropriate inhibitor by one-way t-test. Statistical analyses were performed using Prism 7.0 and 8.0 (Graphpad, USA).

## 3. Results

### 3.1 Phenotype of the differentiated brain endothelial-like cells

Hydrogel cultures of the blood-brain barrier model were set up as indicated in Fig 1 with primary human astrocytes incorporated into the collagen hydrogel, and a monolayer of brain endothelium differentiated from hiPSCs. Structural phenotypic markers of brain endothelial cells were measured at day 10 after differentiation by immunofluorescence (Fig 2). At this stage the cells had continuous tight junctions with strong junctional expression of ZO-1, occludin and claudin-5, which is characteristic of brain endothelium. The cells had weak junctional expression of the general endothelial marker VE-cadherin and weak expression of PECAM (CD31), which was not specifically located at junctions.

Further examination of the hydrogel cultures was carried out by TEM (Fig 3A). The endothelial-like cells appear as a monolayer 2–3μm deep, with tight junctions. Astrocyte processes were associated with the basal membrane of the endothelium in some areas. The cells also had structures corresponding to Weibel Palade bodies, which were originally described as bundles of fine tubules approximately 0.1μm thick and up to 3μm in length [22]. However, it was notable in our differentiated hiPSCs that these structures were in the perinuclear region and lacked the electron dense membrane of mature Weibel Palade bodies (Fig 3B), which would normally be distributed throughout the cytoplasm. Immunofluorescence confirmed that the differentiated endothelial-like cells do express von Willebrand factor (Fig 2) which is a stored component of the Weibel Palade body. The TEM images also showed that the differentiated cells expressed large lipid vesicles and included some electron dense granular material (Fig 3A). Both components are seen in the undifferentiated hiPSCs and appear to be progressively lost during differentiation (S1 Fig). Double membrane vesicles (Fig 3A) were also seen in many cells. This feature is characteristic of autophagosomes, and it suggests that components of the hiPSCs are being broken down as part of the differentiation process.

At day-10 after switching to UC medium it appears that the differentiated cells have acquired key markers of brain endothelium (claudin-5, VE-cadherin, PECAM, VWF). However, at this time the cells are thicker than primary brain endothelium, have some residual components of the hiPSCs and their Weibel Palade bodies are immature. This data indicates that the cells have differentiated but the process is still ongoing.

### 3.2 Trans endothelial electrical resistance and paracellular permeability

Transendothelial electrical resistance (TEER) of the cells on hydrogels or on filters alone was measured using a Cellzscope starting at day 10 after differentiation and extending for up to 10 days after establishment of the hydrogel cultures. Cells on filters alone, initially had typical TEER values in the range 1000–3000 $\Omega.cm^2$ which was maintained for 3 days after the cultures were established. It then declined progressively up to day 10, at which time the TEER was 300–800 $\Omega.cm^2$ (S2 Fig). The TEER values for hydrogels were compared with values for matched cell cultures on filters only. One representative example of Cellzscope data is shown in Fig 4. During the first 48hours, the TEER of the hydrogels was in the range 300–2000 $\Omega.cm^2$ –ie

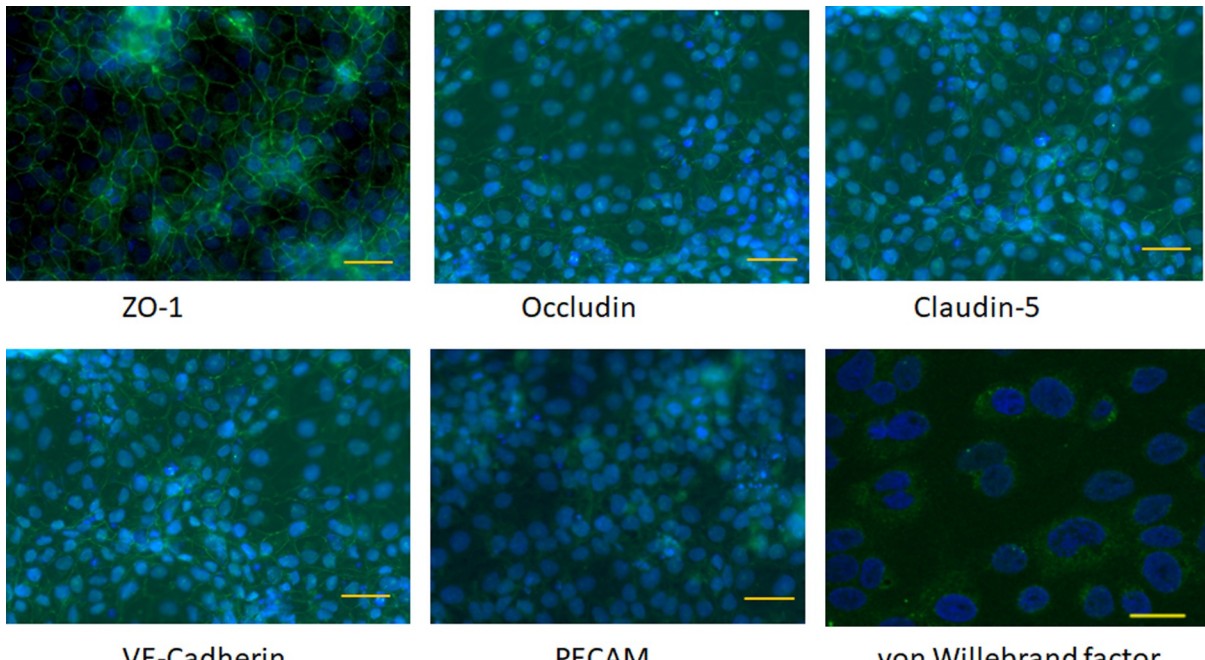

**Fig 2. Immunofluorescence micrographs of differentiated (d10) endothelial-like cells showing junctional expression of ZO-1, occludin, claudin-5 and VE-cadherin, weak surface expression of PECAM and granular expression of von Willebrand factor in the cytoplasm (green).** The cell nuclei were stained with DAPI (blue). Scale bars = 50μm. Scale bar on VWF = 20μm.

considerably lower than the TEER on filters alone. Collated TEER measurements taken at 48 hours after seeding of the hydrogels by Endohm meter confirmed that TEER values for hydrogels were significantly lower than for equivalent cells on filters alone (Table 2).

The data imply that the gels have negligible resistance, as they are similar to filters alone. But endothelial-like cells on the hydrogels have lower resistance than cells on filters alone. This could be due to an intrinsic property of the cells on the hydrogels and/or due to the different geometry of the two systems. In hydrogels the diffusion pathway for ions between the junctions in the endothelial monolayer and the holes in the filter is less tortuous than the diffusion path on filters alone; the narrow gap between the basal membrane of the cells and the surface of the filter restricts molecular movement.

Following the observations on TEER, it was decided to carry out all the transport assays at 24–72 hours after seeding of the hydrogel cultures, when TEER values were highest (Figs 4 and S2). Moreover, all hydrogel cultures were measured by Endohm meter before and after transport assays and only cultures that maintained TEER >200 $\Omega$.cm$^2$ at both the start and end of the assay were included as valid results. It was found that TEER values declined over the course of a 6hr transport assay (Table 3), but more than 90% of cultures maintained an acceptable TEER over this time.

The permeability of the hydrogel cultures for two paracellular fluorescent tracers, was compared with cultures on filters alone. Lucifer yellow (Ly, 443 Da, Stokes radius 0.5nm) and 70kDa FITC-dextran (Dx, hydrodynamic radius 7.2nm), have often been used to estimate the tightness of endothelial junctions, in relation to the pore pathway and leak pathway through the junction. The rate of transfer of the tracers to the lower chamber was measured in hydrogel cultures. No transfer could be detected in the first hour, presumably due to the length of the diffusion pathway across the hydrogel from the basal layer of the endothelium to the filter. The transport rate was derived between 2–6 hours. Examples are shown in S3 Fig. Permeability, Pe,

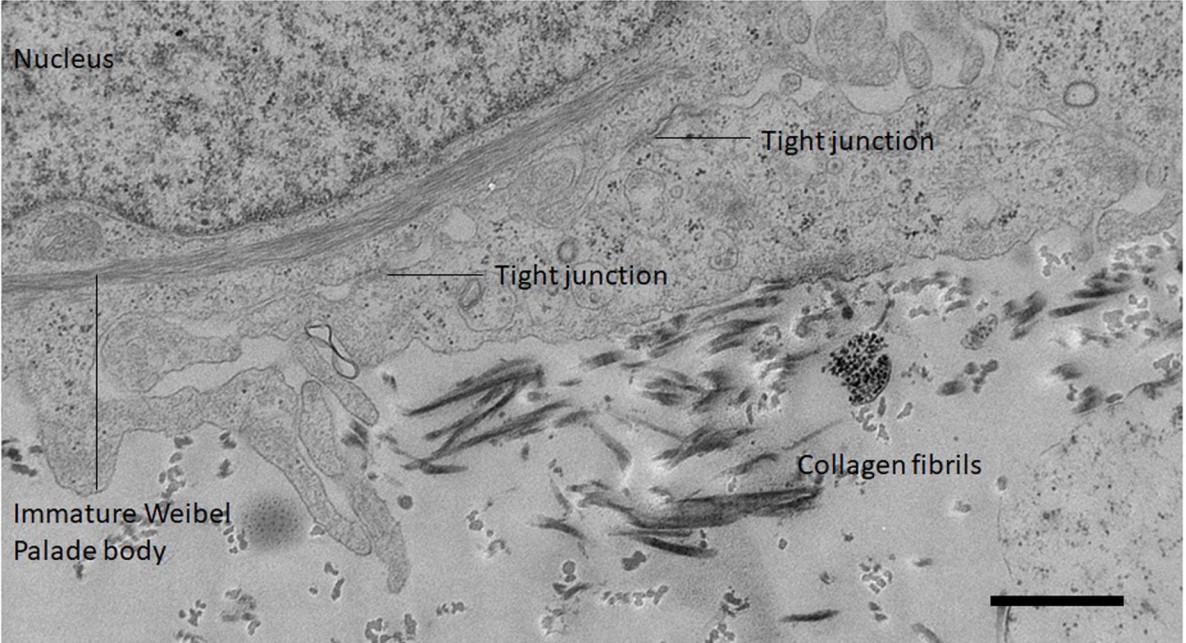

**Fig 3. Transmission electron microscopy images of astrocytes in a collagen hydrogel with differentiated endothelial-like cells on the surface.** A. A transverse section with the apical surface of the endothelium at the top. B. Transverse section showing the basal surface of the endothelial cells. Scale bars = 1µm.

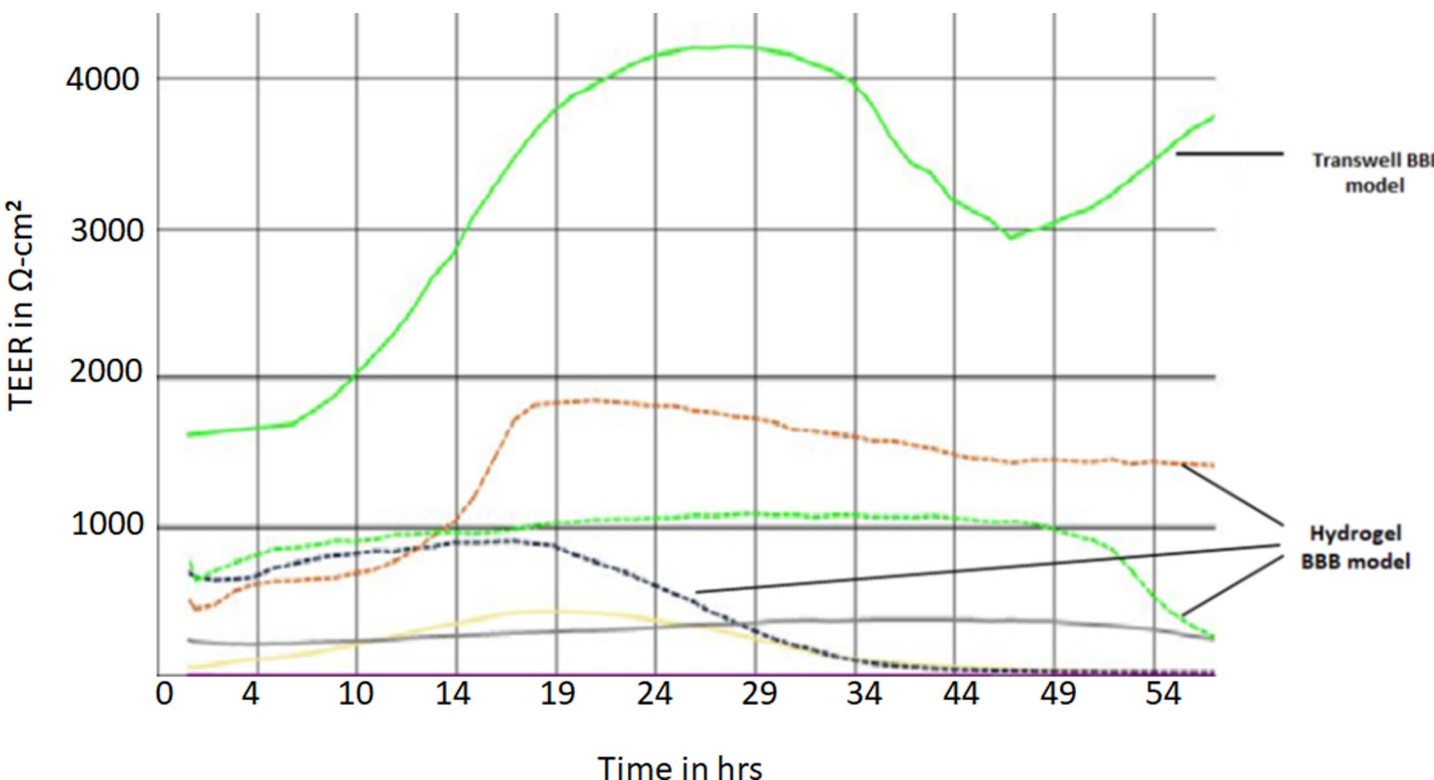

**Fig 4. Trans endothelial electrical resistance of 5 hydrogel cultures and 1 matched culture on 0.4μm pore-size filter only (no gel) measured for 54hrs after culture set up, using a Cellzscope.** Each trace represents the resistance from 1 hydrogel or transwell, measured at 1 minute intervals.

was calculated, in comparison with filter alone or filter+gel (no cells) as appropriate [20]. The results shown in Table 4 imply that permeability for 70kDa FITC-dextran was similar for hydrogels and matched cultures on filters whereas permeability for lucifer yellow was much higher on hydrogels than on filters only. It was also noted that individual cultures which had a high TEER value at the start of the assay also had a relatively high permeability for lucifer yellow. This suggests that TEER can be used as a proxy measurement for the permeability of a low molecular weight tracer.

### 3.3 Transport function of transferrin receptor

The transferrin receptor (TFR1) has been previously used as a target for receptor-mediated transport of large biomolecules across brain endothelium [23]. It was important therefore to establish whether this receptor was functionally active in the hydrogel cultures. Three

**Table 2. TEER values of differentiated cultures.**

|  | Cultures (n) | TEER $\Omega.cm^2$ * |
|---|---|---|
| Cells on hydrogel +filter | 31 | 787 ± 57 |
| Hydrogel + filter (no cells) | 11 | 44 ± 3 |
| Cells on filter | 8 | 1399 ± 127 |
| Filter alone | 9 | 41 ± 4 |

TEER measured at day 10 after differentiation = day 2 after seeding the hydrogels or filters.
* Mean ± SEM

**Table 3. TEER values of hydrogel BBB models during transport assays.**

| Time | TEER $\Omega.cm^2$ * |
|---|---|
| 0hr | 787 ± 57 |
| 6hr | 496 ± 40 |

* Mean ± SEM of n = 31 hydrogels

fluoresceinated polypeptides which bind to the receptor were compared with a control peptide for their ability to cross from the apical to basal chamber of the hydrogel cultures. Pep1, Pep-10 and Pep-10M, have been shown to cross hCMEC-D3 cells. Pep-2 also binds the TfR but was less effective at transcytosis. Pep-R1 which does not bind the receptor, was used as control [19]. Examples of transport curves are shown in S4 Fig. Transport was linear between 2hrs and 6hrs for all peptides (Mol wts 1.5–2.5 kDa). The total peptide accumulated in the lower chamber of the hydrogels at the end of the assay is shown in Fig 5. All four TfR-binding peptides were transported at a significantly higher rate than the control peptide, which indicates that the uptake and transport by the TfR is active.

## 3.4 Activity of multi-drug resistance transporters

An important function of the blood-brain barrier is the activity of the multi-drug resistance transporters ABCB1 (Pgp) and ABCG2 (BCRP), in preventing a wide range of xenobiotics and therapeutic drugs from entering the CNS. The functionality of ABCB1 was tested on hydrogel cultures, with the fluorescent substrate rhodamine-123 (rho-123) in the presence or absence of its specific inhibitor zosuquidar. The results of a representative experiment are shown in Fig 6. In the presence of zosuquidar, significantly more rho-123 accumulated in the basal chamber (P = 0.0024), indicating that ABCB1 was functionally active.

The functionality of ABCG2 was also tested in hydrogel cultures, using the ABCG2 substrate mitoxantrone in the presence or absence of its inhibitor Ko143. In this case, mitoxantrone could only be detected in the lower chamber after 24hrs. The results were variable with some experiments showing increased transfer in the presence of the inhibitor and some showing no significant difference. To clarify this issue, the assays were repeated with cell-cultures on filters alone (no hydrogel). Again, the results were variable. A typical assay is shown in Fig 6. The inhibitor increased the rate of transfer, but due to the high variability between individual cultures, this was not significantly different from cultures without inhibitor (P = 0.062, by one way t-test). These results suggest that ABCB1 is present and functional in the cultures, whereas ABCG2 is expressed at the level of mRNA (see below), but functional expression is still developing.

## 3.5 mRNA expression profile of differentiated endothelial-like cells

RNA was isolated from cells at day 8 after the start of differentiation. RNA was also isolated from the adherent cell fraction (which was then used in the hydrogels), the non-adherent and

**Table 4. Permeability of hydrogel cultures to paracellular tracers.**

| Tracer | Cell culture on: | Pe (x10$^{-5}$) cm.s$^{-1}$ |
|---|---|---|
| 70kDa FITC-dextran | Hydrogel + filter (Filter only) | 10.68 ± 0.85 * (14.9 ± 8.3) |
| Lucifer yellow 433Da | Hydrogel + filter (Filter only) | 455 ± 109 (23.1 ± 14.5) |

* Mean and SEM from 4 independent hydrogel cultures.

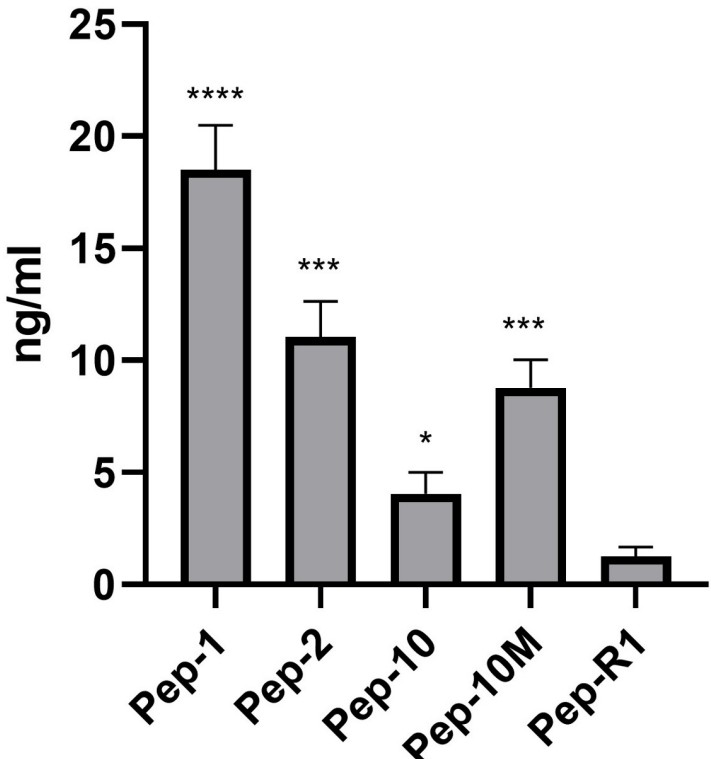

**Fig 5. Transcytosis of TfR-binding peptides to the lower chamber of hydrogel cultures at 6 hours after application to the apical compartment.** Values are expressed as the concentration in the basal chamber, combined data from 2 independent experiments (n = 7) determined by fluorimetry. Data was analysed by one way ANOVAR using Welche's correction for different SD values (P< 0.0001). Subsequent analysis compared TfR binding peptides with Pep-R1 by Tukey's test. **** P<0.0001, *** P< 0.001, * P<0.05.

unfractionated cells of day 8. The RNA was applied for high-throughput qPCR measurements and the genes described below were analysed in detail–tight junction markers (CLDN1, CLDN3, CLDN5, OCLN, TJP1 (ZO-1)), endothelial markers (VWF, CDH5 (VE-cadherin)), receptors, transporters (LRP1, TFR1, SLC2A1 (Glut1)) and multi-drug resistance transporters (ABCG2 (BCRP), ABCC1, ABCC2, ABCC3, ABCC4, ABCC5) Expression was normalised against PPIA in all cases. There was considerable variation in the levels of expression between the different RNA preparations and in respect of these genes, there was no consistent significant difference in expression between the unfractionated differentiated cells and the adherent cells used in the hydrogels. Expression was then compared between the pre-differentiated hiPSCs and the differentiated cells (S5 Fig). The results were in accordance with the immunofluorescence data (Fig 2) and showed an increase in TJP1, OCLN, CLDN3 and CLDN5 following differentiation, but a decrease in CLDN1. This data reinforces the view that both the hiPSCs and the differentiated cells have tight junctions, but the claudin composition of the junctions changes during differentiation towards the brain endothelial phenotype. There was a strong increase in LRP1 and SLCA2A1 expression, but only small increases in TFR1, CDH5 and VWF mRNA.

All of the multidrug resistance transporters increased during differentiation, and the profile of expression of the ABCC transporters is similar in the differentiated endothelial-like cells to that previously reported in human brain capillaries *in situ* [24].

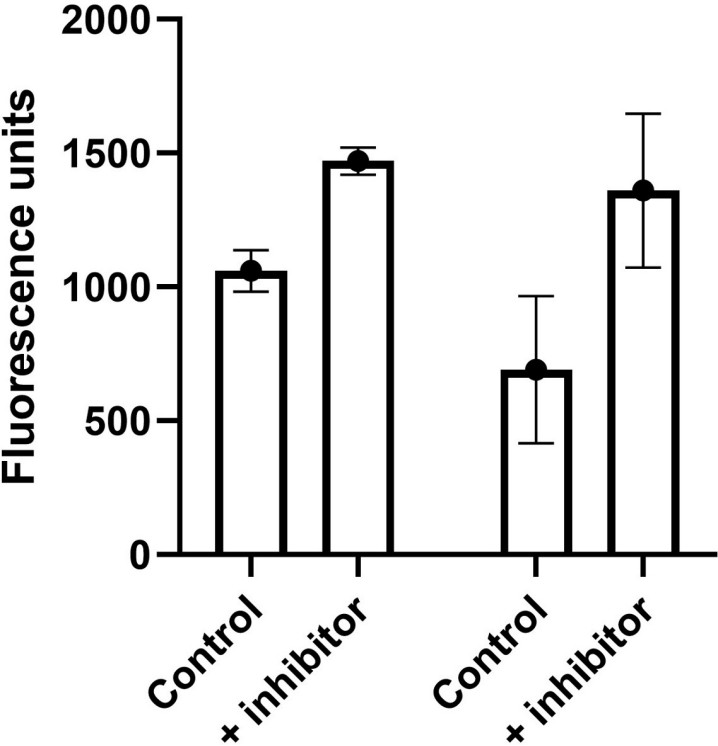

**Fig 6. Activity of ABCB1 measured with Rho-123, comparing transfer across hydrogel cultures at 6hrs, in the absence or presence of the inhibitor zosuquidar.** Mean ± SEM, n = 4, P = 0.0024 by one way t-test. Activity of ABCG2 measured with mitoxantrone, comparing transport at 24hrs across cells on filters, in the absence or presence of its inhibitor Ko143. Mean ± SEM, n = 6, P = 0.062.

## 4. Discussion

The aim of this study was to adapt a hydrogel model of the human blood-brain barrier [17] by replacing the immortalised brain endothelial cell line hCMEC/D3, with endothelial-like cells differentiated from hiPSCs [15, 25].

Two modifications of the previously-described method were found to be essential to successfully adapt the model. Firstly, the concentration of type-1 collagen in the hydrogel was increased from 1mg/ml to 5mg/ml. This increased the rigidity of the hydrogel, providing a better substrate for the endothelium. Previous studies have shown that a concentration of 3mg/ml collagen is inadequate to maintain the differentiated endothelium in a microvessel model and 7mg/ml collagen cross-linked with genipin has been recommended to model the stiffness of the brain in vivo [26]. However, the use of genipin presents potential problems with cytotoxicity and a concentration of 5-7mg/ml collagen was adequate to produce a barrier to paracellular tracers and high TEER [26, 27]. The integrity of the barrier is also dependent on the surface substrate and the fibronectin/collagen type IV substratum was essential for spreading of the endothelial monolayer and barrier properties [28]. The high collagen concentration in the hydrogel (5mg/ml) also has the advantage that it resists contraction, when astrocytes are incorporated in the gel, which is a problem with lower collagen concentrations. However, viability of astrocytes in the high concentration gels appears to be lower over time, possibly due to poorer diffusion of nutrients and oxygen in stiffer gels.

The second modification of the protocol used by other members of the Im2pact consortium [15], was preselection of high adherence cells. Without this step, it was found that the

endothelial monolayers sometimes lost small areas of cells, and the cultures were not usable. It appeared that the endothelial-like cells were still heterogeneous at day 8 of differentiation and since hydrogels are not as stiff as filters, the cells are more easily dislodged or fail to cover some patches of the gel surface, hence the requirement to use high-adherence cells. Adhesion of hiPSCs depends partly on α5 integrins and their binding to fibronectin [29], whereas collagen receptors (α1 and α2 integrins) are more important for endothelial cell adhesion to type-IV collagen and development of endothelial adherens junctions [30]. One can therefore infer that changes in integrin expression would be expected as these cells differentiate, and cells that have not fully differentiated would adhere less-well to the type-IV collagen coat on the hydrogel, than primary endothelium.

It has been debated as to whether other cell types may be needed in the hydrogels in order to develop the optimum phenotype of the endothelial cells. We did not observe any difference in cultures with or without astrocytes in the hydrogels, and others have reported that barrier function is not significantly affected by astrocytes or pericytes in the gels [31], although the combined presence of differentiated neural cells with pericytes and astrocytes (triculture) beneath the endothelium did significantly increase TEER and SLC2A1 mRNA in comparison with endothelium alone in a Transwell® model using differentiated hiPSCs [28]. The astrocytes used in the model described here do not appear to contribute to the barrier properties of the endothelial-like cells. However the study does confirm that astrocytes can still be incorporated into the hydrogels, with the endothelial-like cells forming the barrier. Astrocytes or other glial cells would only be incorporated if the aim of an assay was to determine whether molecules or cells that have crossed the endothelium then interact with glia; cells in the hydrogels can be recovered by collagenase digestion to investigate changes to their phenotype or uptake of transported molecules.

In agreement with previous work [26] we found that the TEER values of the endothelial-like cells peaked at 48 hours after the cultures were set up and then declined gradually over 10 days. The media changes, required during this period, produced transient drops in the TEER value lasting for 1–2 hours. The TEER of the cultures fully recovered by 3 hours after the medium change. Also, in agreement with previous work [27] TEER values on hydrogels were 20–30% of the equivalent value of the same cells on filters alone. It is uncertain whether the lower TEER values on hydrogels are due to weaker tight junctions of endothelium on the gels or to the more open diffusion pathway from the basal membrane of the endothelium to the lower chamber of the transwell. It has been proposed that a minimum TEER of 150–200$\Omega$.cm$^2$ is required for any BBB model that is used to measure transport of small therapeutic drugs [32]. In our study we chose 200$\Omega$.cm$^2$ as the cut-off value for acceptable cultures. Generally, more than 90% of hydrogel cultures had a TEER >200$\Omega$.cm$^2$. We noted that individual time-point values of TEER on the Endohm meter corresponded well with the values from the Cellzscope. Since many laboratories do not have access to a Cellzscope, we recommend checking before and after any transport assay that TEER values are >200$\Omega$.cm$^2$, to confirm that the endothelial monolayer has not been disrupted by the assay reagents or protocol.

An important element of any blood-brain barrier model is the presence and activity of transporters. The multi-drug resistance transporter ABCB1 was present and functionally active as expected. ABCG2 was present but functional activity was variable between cultures. The profile of multi-drug resistance genes (ABCC1, 3, 4, 5) corresponded with that previously reported for brain endothelium [2, 12]. The potential for using the transferrin receptor as a target for transport of larger molecules was demonstrated by the transcytosis of TfR-binding peptides. The endothelial-like cells also strongly expressed LRP1 and could therefore potentially be targeted for drug delivery tests using e.g. angiopeptide Ang-2 [33].

Models of the blood-brain barrier based on differentiated hiPSCs are increasingly used in research and drug development. The models may contain several other cell types [31, 34] and hydrogel models can readily be applied to mixed cell cultures, since glia or neurons can be incorporated into the gel. The differentiated endothelium is reported to be relatively stable on hydrogels and in hydrogels [27, 35]. However, some studies have failed to find some key brain endothelial markers in the differentiated hiPSCs and have identified some epithelial markers [36]. The explanation for the discrepancies between laboratories appears to lie in what time-point in the differentiation is chosen and whether the cells are assayed for mRNA, expressed protein, functional protein or correctly localised protein.

The electron microscopy carried out in our study indicates that the endothelium is not fully differentiated at the time when endothelial markers are first detectable. It has immature Wei-bel Palade bodies, is thicker than primary brain endothelium and expresses residual features of the hiPSCs, including lipid vesicles and electron-dense granular material (Fig 3A and 3B). This finding is reinforced by the immunofluorescence studies (Fig 2) which show relatively low lev-els of VE-cadherin and PECAM at this time. It has often been assumed that the hiPSCs have differentiated into brain endothelium when characteristic markers are detectable (claudin-5, von Willebrand factor, TFR1, VE-cadherin etc). However, it should be cautioned that some of the tight junction markers seen on brain endothelium (occludin, ZO-1) are also present on the hiPSCs. Therefore tight-junction markers and high TEER, although desirable in a BBB model, do not indicate that the cells have necessarily differentiated into endothelium. Indeed, there is a rationale for leaving cells for a few days in the hydrogel culture after differentiation to allow more time to increase functional expression of brain endothelial elements and lose hiPSC components. In comparison with the original hydrogel model with hCMEC/D3 cells [17], the new model shows greatly improved barrier properties, which allow transport studies of smaller molecules and substrates of the multi-drug transporters. In the new model, measurement of transendothelial transport of smaller molecules is not confounded by paracellular leakage. One disadvantage is that preparation of the model takes an extended period of time, but this is pri-marily due to the time needed to differentiate the endothelial-like cells. This process could be abbreviated by the use of frozen cells from a large batch [37].

## 5. Conclusion

The hydrogel model of the blood-brain barrier using endothelium differentiated from hiPSCs is relatively easy to set up and allows access to apical and basal surfaces for transport studies. Glia can be incorporated into the gel matrix. The TEER is lower and paracellular permeability slightly higher than for equivalent cultures on transwell filters alone. Multi-drug resistance genes are present and ABCB1 was shown to be functionally active. The transferrin receptor and LDL receptor LRP1 are strongly expressed and transcytosis, mediated via TFR1, was dem-onstrated using TfR-binding peptides. The model is usable for up to 7 days after set-up.

## Supporting information

**S1 Fig. Transmission electron microscopy image of hiPSC on matrigel.** Transverse section showing the lipid vesicles and electron-dense granular material, which is progressively lost as the cells differentiate into endothelium. Scale bar = 1μm.
(TIF)

**S2 Fig. Trans endothelial electrical resistance (TEER) of differentiated endothelial-like cells on filters, measured by Cellzscope over 220hours.** Medium changes were made at 72hrs and 170hrs. A and B, cultures with antibiotic (Penicillin/streptomycin): C and D, cultures

without antibiotic. E and F, Caco-2 cells, cultured in parallel.
(TIF)

**S3 Fig. Examples illustrating the accumulation of paracellular tracers (Lucifer yellow and 70kDa dextran) in the basal chamber of hydrogel BBB cultures compared with the hydrogel+ filter only.** Matched cells on transwell filters only and transwell filter only are also shown. The background fluorescence released from the hydrogels is show as 'medium background' ie with no tracer. This type of plot was used to derive the permeability values of the paracellular tracers.
(TIF)

**S4 Fig. Examples illustrating the accumulation of TfR-binding peptides in the basal chamber of the hydrogel BBB model compared with hydrogels only (no cells).** Matched endothelial cells on transwells (Cells only) compared with transwell filter only are also shown.
(TIF)

**S5 Fig. Expression of brain endothelial cell markers in differentiated endothelium.** Expression in the differentiated endothelium was measured by qPCR and the level of expression compared with pre-differentiated hiPSCs (= 1). Values show mean and standard deviation from 4 preparations.
(TIF)

## Acknowledgments

Dr Moriah Katt gave important advice on the required composition of the hydrogel matrix required to support the IPSC-derived endothelium.

## Author Contributions

**Data curation:** Radka Gromnicova, Andreas Brachner.

**Formal analysis:** Nandita Rahatekar Singh, Andreas Brachner, Winfried Neuhaus, David Male.

**Funding acquisition:** Ignacio A. Romero, Winfried Neuhaus, David Male.

**Investigation:** Nandita Rahatekar Singh, Radka Gromnicova, Andreas Brachner, Igor Kraev, Winfried Neuhaus, David Male.

**Methodology:** Nandita Rahatekar Singh, Radka Gromnicova, Igor Kraev, David Male.

**Project administration:** Ignacio A. Romero, Winfried Neuhaus, David Male.

**Supervision:** Ignacio A. Romero, Winfried Neuhaus, David Male.

**Writing – original draft:** Nandita Rahatekar Singh, David Male.

**Writing – review & editing:** Nandita Rahatekar Singh, Igor Kraev, Ignacio A. Romero, Winfried Neuhaus, David Male.

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
