## [Decision Letter · Decision Letter 0]

10 Jan 2023

PONE-D-22-33370A hydrogel model of the human blood-brain barrier using differentiated stem cellsPLOS ONE

Dear Dr. Male,

Thank you for submitting your manuscript to PLOS ONE. After careful consideration, we feel that it has merit but does not fully meet PLOS ONE’s publication criteria as it currently stands. Therefore, we invite you to submit a revised version of the manuscript that addresses all the points raised during the review process.

Two reviewers have evaluated the manuscript and found it interesting. They asked for amendments including, among others, reorganization of the figures, more detailed description of the methods and discussion about the advantages and potential use of the new model with clear description of the novelties.==============================

We look forward to receiving your revised manuscript.

Kind regards,

Mária A. Deli, M.D., Ph.D.

Academic Editor

PLOS ONE

Journal Requirements:

Reviewers' comments:

Reviewer's Responses to Questions

**Comments to the Author**

1. Is the manuscript technically sound, and do the data support the conclusions?

Reviewer #1: Partly

Reviewer #2: Yes

2. Has the statistical analysis been performed appropriately and rigorously? 

Reviewer #1: No

Reviewer #2: Yes

3. Have the authors made all data underlying the findings in their manuscript fully available?

Reviewer #1: Yes

Reviewer #2: Yes

4. Is the manuscript presented in an intelligible fashion and written in standard English?

Reviewer #1: Yes

Reviewer #2: Yes

5. Review Comments to the Author

Reviewer #1: Singh et al. developed and characterized a novel human blood-brain barrier (BBB) model based on a collagen hydrogel containing astrocytes and endothelial cell monolayer from human inducible pluripotent stem cells (hiPSCs). The manuscript is an important area of BBB and CNS research. Human induced pluripotent stem cells (hiPSCs) have emerged as a promising source of reproducible human-derived biological material, showing great potential for the development of robust BBB in vitro models. This study convincingly demonstrates it and provides a protocol, which is useful for the development of strategies for therapeutic intervention against various neurological disorders. The proposed model can be also useful for other researchers in the field to study the transport of small therapeutic drugs through the BBB in vitro.

Major comments:

1. Fig. 1 is very limited and needs to be improved to increase the reproducibility for the reader (especially the part related to the differentiation of human iPSCs to endothelial-like cells). The key model components are listed to the right of the Transwell system; however, this overview of the primary components is not accompanied by arrows to point out their placement (please see the following papers with detailed workflow examples: doi: 10.3791/61629.; doi: 10.1080/10837450.2021.1872624). All used cell types should be visualized. Besides, the Figure 1 legend should include the description of critical steps as well as full names for abbreviations used in this Figure.

2. In the Results, the authors made the following statement: "The cells had weak junctional expression of the general endothelial marker VE-cadherin and weak expression of PECAM (CD31)…'' (p. 15, lines 4-6 from the top). How the "weakness" of the fluorescence signal coming from CD31 was measured? There was no quantitative analysis performed for this conclusion. Besides, the authors did not indicate whether the immunofluorescence staining of cultured cells was performed on glass coverslips or permeable Transwell inserts. If the latter applies, the insert membrane material (polycarbonate or polyester) can play a significant role in the visualization of junctional proteins (please see the paper doi: 10.12659/msmbr.900656.) and therefore could potentially explain why CD31 staining looks not as sharp as the other obtained images.

3. In the Results, the authors made the following statement: "The cells also had structures corresponding to Weibel Palade bodies, which appear in TEM images as bundles of fine tubules approximately 0.1μm thick and up to 3μm in length [22]." (p. 15, lines 8-10 from the top). Did the authors measure these parameters or these numbers are coming from reference #22? There should be an indication of software or technique used for the quantification of morphological structures on TEM images.

4. The representation of information in Figures and Tables needs a revision. Too many "short" figures may obscure the important results. It would be better to group all TEER data (measured with both cellZscope and Endohm meter) and paracellular permeability into a multi-panel figure (those with parts A, B, C, D, etc.). The same strategy can be applied to combining Figures 3 and 4 into a multi-panel figure. Therefore, Figures S4.1 – S4.4 should be combined as one main figure for part 3.3 of the Results. Similarly, Suppl. Figures S5.1-S5.4 should be combined and represented as the main figure for part 3.5. Parts 3.3 and 3.5 of the Results do not have any main figures, which makes the reader question what makes these findings less important. Transforming Tables 2 and 5 into bar graphs would be a much better visualization for comparing values between groups of data. Table 4, in turn, can be transformed into a Supplementary Figure.

5. It would be more appropriate to create a subsection entitled "Statistical Analysis" for the Materials and Methods instead of quite specific "Data analysis of permeability and transport assays". Does this mean the other data (qPCR and transporter activity) did not undergo statistical analysis? Why did authors use "one way ANOVA followed if appropriate by unpaired t-test" for data comparison instead of a one-way ANOVA followed by Tukey's post-hoc test?

6. It looks like Fig. 7, Fig. S3, and Fig. S4 have only one sample per group or each time point. The sample size should be increased to run a statistical analysis or make the conclusions.

7. The limitations of the developed model should be added to the Discussion.

Minor comments:

1. In the Introduction, the authors said that "Hence, one desirable characteristic of a BBB model is the expression of continuous tight junctions, a high TEER and corresponding low permeability for paracellular tracers". It would be better to say "characteristics" since these are different BBB model aspects.

2. It is not very clear why BBB models need the incorporation of a hydrogel. It should be better described in the Introduction.

3. What is the advantage of using astrocytes in the developed BBB model? Why the manuscript title lacks "astrocytes"?

4. Part 2.6. "Characterization of brain endothelial-like cells derived from human- induced pluripotent cells" should be replaced with a more specific title, for example, "Immunofluorescence staining". What kind of microscopy was used for Immunofluorescence staining imaging?

5. P. 22, line 14: "The second modification of the basic protocol was preselection of high adherence cells.". Did the authors mean their previously published protocol?

6. Please examine if all used abbreviatures in the text are kept consistently. For example, Line 3 in 2.2. Cell selection before seeding states "On day 8 after shift to UC medium, the cells were detached with accutase…". Did you mean UM? The previous paragraph states that "the cells were shifted to unconditioned media (UM) for 6 days".

7. Please revise the manuscript for minor typographical and grammatical errors. For example, the dots are missing between the sentences at line 15 of page 3 (Introduction) and line 9, of page 6 (Part 2.1).

8. All images presented in Fig. 2 have two colors. Does the blue color in the nucleus represent DAPI staining? Please specify colors.

9. Supplementary Fig. 1 has no arrows pointing out individual morphological structures or a scale bar.

10. Sample size (n) and data expression such as Mean ± SEM need to be added to the figures with TEER and transport assay data.

11. Please add lines with asterisks to denote statistical significance on Suppl. Figures S5.1-S5.4.

Reviewer #2: The manuscript by Singh and colleagues deals with the description of the establishment of a hydrogel in vitro blood brain barrier (BBB) using hIPSC derived endothelial like cells. While the paper does highlight some new data, the authors should consider strengthening their discussion of how their work with hydrogel stands out in contrast to other studies and also for which purposes the hydrogel model can be used for. Furthermore, the rationale for using a hydrogel model should be better described. Additionally, there are serious concerns regarding experimental design and data interpretation that preclude publication of this manuscript in its present form. There are some other points mentioned below that the authors may want to consider:

1. There are several publications in the scientific literature that describe the use of hIPSCs derived brain endothelial like cells. Based on the authors' data, it is not clear as to how their model system is more advantageous or more novel than these other published in vitro models utilizing hIPSCs derived brain endothelial like cells.

2. Indeed, it is well known that other cell types of the neurovascular unit (i.e., astrocytes) can regulate the endothelium phenotypes in terms of barrier properties and transporter expression regulation. The authors described the use of astrocyte. However, the “general beneficial” effects of astrocytes on the barrier model are not described or discussed. In addition, the astrocytes have not been characterized, and the source of them is missing in the method section. Could you discuss on howthe astrocytes potentially affect the monolayers? It has been used but why this specific cell type and how could it affect the cultures?

3. The authors should re-consider the permeability calculations. From the figures in the supplementary, it can be seen that the steady-state fluxes were not constant during the time course of the transport experiment. The authors should consider adding a statement whether sink conditions were maintained, and also provide the equation for calculating the permeability.

4. The protein expression levels of key targets identified in the study would add more valuable information to the functional aspects of these BBB markers or transporters assessed in the hydrogel model. That information may be a great addition to the current study as the authors are comparing different culture configurations; transwell vs. hydrogel.

6. PLOS authors have the option to publish the peer review history of their article (what does this mean?). If published, this will include your full peer review and any attached files.

Reviewer #1: **Yes: **Nikolai Fattakhov

Reviewer #2: No

---

## [Author Response · Author response to Decision Letter 0]

15 Feb 2023

We have checked the journal requirements

Also we have deleted the short section on antibody transport (previously 'data not shown') since it was not material to the focus of the paper.

---

## [Decision Letter · Decision Letter 1]

21 Mar 2023

A hydrogel model of the human blood-brain barrier using differentiated stem cells

PONE-D-22-33370R1

Dear Dr. Male,

We’re pleased to inform you that your manuscript has been judged scientifically suitable for publication and will be formally accepted for publication once it meets all outstanding technical requirements.

Kind regards,

Mária A. Deli, M.D., Ph.D.

Academic Editor

PLOS ONE

Additional Editor Comments (optional):

Reviewers' comments:

Reviewer's Responses to Questions

**Comments to the Author**

1. If the authors have adequately addressed your comments raised in a previous round of review and you feel that this manuscript is now acceptable for publication, you may indicate that here to bypass the “Comments to the Author” section, enter your conflict of interest statement in the “Confidential to Editor” section, and submit your "Accept" recommendation.

Reviewer #1: All comments have been addressed

2. Is the manuscript technically sound, and do the data support the conclusions?

Reviewer #1: Yes

3. Has the statistical analysis been performed appropriately and rigorously? 

Reviewer #1: Yes

4. Have the authors made all data underlying the findings in their manuscript fully available?

Reviewer #1: Yes

5. Is the manuscript presented in an intelligible fashion and written in standard English?

Reviewer #1: Yes

6. Review Comments to the Author

Reviewer #1: I am satisfied with the author’s responses to my comments. Needed corrections and clarifications have been made. Revised manuscript is easier to follow. The figures are better organized as well. I recommend that the revised paper be accepted.

7. PLOS authors have the option to publish the peer review history of their article (what does this mean?). If published, this will include your full peer review and any attached files.

Reviewer #1: **Yes: **Nikolai Fattakhov

---

## [Editor Report · Acceptance letter]

27 Mar 2023

PONE-D-22-33370R1 

A hydrogel model of the human blood-brain barrier using differentiated stem cells 

Dear Dr. Male:

I'm pleased to inform you that your manuscript has been deemed suitable for publication in PLOS ONE. Congratulations! Your manuscript is now with our production department. 

Kind regards, 

on behalf of

Prof. Mária A. Deli 

Academic Editor

PLOS ONE